# Micronutrient Intake Adequacy in Men and Women with a Healthy Japanese Dietary Pattern

**DOI:** 10.3390/nu12010006

**Published:** 2019-12-18

**Authors:** Tomoko Ito, Kumpei Tanisawa, Ryoko Kawakami, Chiyoko Usui, Kaori Ishii, Katsuhiko Suzuki, Shizuo Sakamoto, Isao Muraoka, Koichiro Oka, Mitsuru Higuchi

**Affiliations:** 1Faculty of Sport Sciences, Waseda University, Saitama 359-1192, Japan; 2Institute of Advanced Active Aging Research, Waseda University, Saitama 359-1192, Japan

**Keywords:** Japanese dietary pattern, micronutrient, dietary reference intakes

## Abstract

This study examined the relationship between a healthy Japanese dietary pattern and micronutrient intake adequacy based on the Dietary Reference Intakes for Japanese 2015 (DRIs-J 2015) in men and women. A cross-sectional study was conducted in 1418 men and 795 women aged 40–87 years, who participated in the Waseda Alumni’s Sports, Exercise, Daily Activity, Sedentariness, and Health Study. Dietary patterns were derived from principal component analysis of the consumption of 52 food and beverage items, which were assessed by a validated brief-type self-administered diet history questionnaire. Micronutrient intakes were quantified using the dietary reference intakes score (DRIs-score) for 21 micronutrients (based on DRIs-J 2015). The healthy dietary pattern score was significantly and positively correlated with the intakes of all 21 micronutrients used for constructing the DRIs-score in men and in women (each, *p* < 0.001). In both sexes, the healthy dietary pattern scores were strongly and positively associated with DRIs-scores (in men: ρ = 0.806, *p* < 0.001; in women: ρ = 0.868, *p* < 0.001), and the DRIs-scores reached a plateau around the highest tertile of the healthy dietary pattern score. These results indicate that a healthy Japanese dietary pattern is associated with adequate micronutrient intakes based on the DRIs-J 2015 in both men and women.

## 1. Introduction

Diet is an important factor for maintaining health and preventing chronic diseases. People consume diets consisting of a variety of foods that provide a combination of multiple nutrients rather than single nutrients or foods, making it important to comprehensively evaluate overall nutrient and food intake in relation to health outcomes. Dietary pattern analysis has been widely used in the field of nutritional epidemiology to evaluate the quantity, variety, or combination of different foods and beverages consumed in a diet [1]. Several studies have shown that dietary patterns identified by principal component analysis are associated with the incidences of various chronic diseases and health indicators [2,3].

The Japanese diet is attracting worldwide attention for its role in contributing to the longevity of the Japanese population [4]. Analyses have shown that the typical Japanese dietary pattern is characterized by high intakes of vegetables, fruits, soy products, mushrooms, seaweed, and fish, and has been referred to as a “Healthy,” “Prudent,” and “Vegetable” dietary-pattern in previous studies [5,6,7,8]. This healthy Japanese dietary pattern is associated with a lower risk of high blood pressure [5], diabetes mellitus [9], cardiovascular disease [10], physical dysfunction [11], and mortality [8], after lifestyle factors such as physical activity and smoking are adjusted for.

Healthy Japanese dietary patterns have been associated with a higher intake of multiple micronutrients [7,8]; therefore, a favorable association between a healthy Japanese dietary pattern and health outcomes may be explained in part by a higher intake of micronutrients. Because micronutrient deficiency conditions are related to many chronic diseases [12], it is important to examine whether dietary patterns meet micronutrient requirements. Several studies have examined whether a healthy Japanese dietary pattern is associated with an adequate nutrient intake based on the Dietary Reference Intakes for Japanese (DRIs-J). The DRIs-J were designed with the aim of preventing lifestyle-related diseases. However, the populations investigated in studies of the association between a healthy Japanese dietary pattern and adequate nutrient intake have been limited to either men and women combined [13] or men only [14], and no studies have investigated women only. Because higher intakes of micronutrients, such as vitamins and minerals, have been associated with a lower risk of breast cancer [15,16] and osteoporosis [7,17] in women, it is important to examine the relationship between dietary patterns and micronutrient intake specifically in women.

Therefore, the purpose of this study was to examine the relationship between a healthy Japanese dietary pattern and the adequacy of micronutrient intakes in men and women, based on the DRIs-J 2015.

## 2. Materials and Methods

### 2.1. Study Design and Participants

The participants included 2242 Japanese men and women aged 40 to 87 years (age criteria: ≥40 years) who participated in the Waseda Alumni’s Sports, Exercise, Daily Activity, Sedentariness, and Health Study (WASEDA’S Health Study), a study whose participants comprised of Waseda University graduates and their spouses between March 2015 and March 2019. All participants provided written informed consent before being enrolled in the study, which was approved by the Ethical Committee of Waseda University (Reference number. 2014-095, 2014-G002). The study was conducted in accordance with the Declaration of Helsinki.

The participants were enrolled in this cohort and answered web-based questionnaires. The participants completed dietary surveys and had anthropometric measurements assessed at 1 of 15 locations across Japan (15 areas: Hokkaido, Gunma, Saitama, Chiba, Tokyo, Kanagawa, Yamanashi, Shizuoka, Aichi, Kyoto, Mie, Osaka, Hyogo, Hiroshima, Fukuoka). We excluded 29 participants from analysis: 18 participants failed to submit the questionnaire on diet, 10 participants met the exclusion criteria of having a dietary energy intake of under 600 kcal or over 4000 kcal, which was determined from the results of their dietary intake questionnaire; and 1 participant had dementia. In total, the results from 2213 participants (1418 men and 795 women) were analyzed. Since we examined the relationship between the Japanese healthy dietary pattern and the adequacy of micronutrient intakes, we did not exclude those with disease from the analysis.

### 2.2. Anthropometry

Height was measured using a stadiometer (YHS-200D, YAGAMI Inc., Nagoya, Japan). Body weight was measured using an electronic scale (MC-980A, Tanita Corp., Tokyo, Japan). Body weight was measured with the participants wearing light clothing and no shoes. The body mass index (BMI) was calculated as the body weight (kg) divided by the square of the body height (m).

### 2.3. Dietary Assessment

The dietary intakes were assessed by a validated brief-type self-administered diet history questionnaire (BDHQ) during the preceding month [18]. We carefully checked all of the answered BDHQs to avoid the effect of misreporting. The BDHQ is a 4-page questionnaire that takes about 15 min to administer. Dietary intakes of 58 food and beverage items, energy, and selected nutrients were estimated using an ad hoc computer algorithm for the BDHQ, based on the Standard Tables of Food Composition in Japan [19]. In previous studies, the validity of dietary intake data (energy, nutrients, and foods) that was assessed using the BDHQ was confirmed using 16-d semi-weighted dietary records as the gold standard [18,20].

### 2.4. Dietary Pattern

The details of the identification of dietary patterns have been described elsewhere [6,14]. In short, we conducted principal component analysis based on energy-adjusted intakes by using a density method of 52 food and beverage items. To determine the number of factors to retain, we considered eigenvalues >1, the Scree test, and the interpretability of factors. Since we aimed to examine the association between a healthy dietary pattern and micronutrient intake, we decided to retain only the first factor that was referred to as a healthy dietary pattern in previous studies [6,7]. The factor scores for individuals were calculated by summing the intakes of the food items weighted by their factor loadings.

### 2.5. Dietary Reference Intakes Score (DRIs-Score)

We used a previously developed DRIs-score [14] to comprehensively assess the adequacy of micronutrient intakes for 21 micronutrients, based on the DRIs-J 2015. Each micronutrient was given a nutrient adequacy score as follows: (1) a score of ‘1’ was allocated if the micronutrient intake level (amount of nutrient intake per 1000 kcal) met or exceeded the recommended dietary allowance (RDA), adequate intake (AI), or tentative dietary goal (DG) (amount of nutrient intake per 1000 kcal) given in the DRIs-J 2015; or (2) a score or ‘0’ was allocated if the micronutrient intake level either did not meet the RDA, AI, or DG, or met or exceeded the tolerable upper intake level. Of the 21 micronutrients, we used the RDA for 13 micronutrients (vitamin A, vitamin B_1_, vitamin B_2_, niacin, vitamin B_6_, vitamin B_12_, folate, vitamin C, calcium, magnesium, iron, zinc, copper), the AI for 6 micronutrients (vitamin D, vitamin E, vitamin K, pantothenic acid, phosphorus, manganese), and the DG for 2 micronutrients (sodium, potassium). Of the 21 micronutrients, we used the tolerable upper intake level for 12 micronutrients (vitamin A, vitamin D, vitamin E, niacin, vitamin B_6_, folic acid, calcium, phosphorus, iron, zinc, copper, manganese). The DRIs-scores were determined by summing the scores of all nutrients to provide a measure of the overall adequacy of micronutrient intake. Thus, the DRIs-score ranged from 0 to 21, with higher scores indicating good balanced micronutrient intake status.

### 2.6. Statistical Analyses

To assess the adequacy of intake for 21 micronutrients, we calculated the proportion of participants with a nutrient adequacy score “1”. To compare characteristics of the participants by sex, we used a Student’s *t* test and Chi-square test. The participants were classified into 3 groups according the tertiles of the healthy dietary pattern score. Because the nutrient intakes are largely affected by sex and age, and an underestimation of micronutrient intake in the individuals with high energy intake may occur when using the nutrient density method, the participants were classified into tertiles by sex and age (40–49, 50–59, and ≥60 years). To assess the trend associations in the characteristics of participants among the tertiles of the healthy dietary pattern, we used linear regression analysis for continuous variables and the Mantel–Haenszel chi-square test for categorical variables, assigning ordinal numbers 1, 2, and 3 to the low, middle, and high tertiles of the healthy dietary pattern, respectively. To assess the relationship between the healthy dietary pattern score and the DRIs-score, we calculated the Spearman’s rank correlation coefficient between the healthy dietary pattern score and the DRIs-score. Nonlinear relationships between variables were visualized by cubic spline curves with 95% confidence intervals.

All measurements and calculated values are presented as a mean ± standard deviation for continuous variables and as percentages for categorical variables. The level of significance was set at a two-sided *p* value < 0.05. We performed all statistical analyses with SPSS, version 25.0 (SPSS Inc., Chicago, IL, USA).

## 3. Results

The results of the principal component analysis identified the major Japanese dietary pattern (Appendix A), which was characterized by higher intakes of vegetables, mushrooms, seaweed, soy products, fruits, and fish compared to the other dietary pattern. This major factor was termed a healthy dietary pattern.

The characteristics of the participants are shown in Table 1. The healthy dietary pattern scores were higher in women than in men (*p* < 0.001), but the DRIs-scores were not significantly different between men and women (*p* = 0.862). Of the 21 micronutrients used for constructing the DRIs-score, the proportion of participants with an adequate intake of vitamin A, vitamin B_1_, vitamin B_2_, sodium, potassium, magnesium, phosphorus, and zinc were higher in women than men, whereas those of vitamin D, folate, vitamin C, iron, and manganese were higher in men than women. The proportion of participants with salt intakes within the DG was very low in both sexes (in men: 0.4%, in women: 1.9%).

The nutrient intakes and the proportions of participants with adequate intakes of micronutrients within each tertile of the healthy dietary pattern score are shown in Table 2 (for men) and Table 3 (for women). The tertile of the healthy dietary pattern score was significantly and positively correlated with the intake of all 21 micronutrients that were used for constructing the DRIs-score in men and in women, respectively (each, *p* < 0.001). There was no significant difference among salt intake in the tertiles of the healthy dietary pattern score in men (*p* for trend = 0.099), but there was a negative association of salt intake between the tertiles of the healthy dietary pattern score in women (*p* for trend = 0.011).

Figure 1 shows the relationships between healthy dietary pattern scores and DRIs-scores by sex. A high healthy dietary pattern score was strongly associated with a high DRIs-score in both sexes (in men: ρ = 0.806, *p* < 0.001; in women: ρ = 0.868, *p* < 0.001). A nonlinear relationship between the healthy dietary pattern score and the DRIs-score was observed in both men and women, and the DRIs-score reached a plateau at around the highest tertile of the healthy dietary pattern score in both men and women.

## 4. Discussion

In the present study, a healthy dietary pattern, characterized by a high consumption of vegetables, mushrooms, seaweed, soy products, fruits, and fish, was identified as the first dietary pattern. Furthermore, a high healthy dietary pattern score was strongly associated with high DRIs-score in both men and women. These results indicate that the healthy dietary pattern is associated with an adequate micronutrient intake in both men and women, based on the DRIs-J 2015.

The characteristics of the healthy dietary pattern identified in the present study were consistent with that of the Healthy, Prudent, and Vegetable dietary patterns identified in previous studies among Japanese populations [5,6,7,8]. This dietary pattern is also similar to the Mediterranean dietary pattern, which is characterized by the consumption of olive oil, vegetables, and fish in the Greek population [21] and the healthy dietary pattern comprised of a high intake of vegetables and fruits in the American population [22]. Similar to our results, studies of Greek [21] and American [22] populations found that healthy dietary pattern scores were higher in women than in men. A study on food choice behaviors in young adults from 23 countries reported that women were more knowledgeable about food and consumed more vegetables and fruits than men [23]. This awareness of food in women may explain the reason why women have higher healthy dietary pattern scores than men.

The DRIs-scores were not significantly different between men and women, although healthy dietary pattern scores were higher in women than in men. In the present study, 94.2% of men and 62.9% of women had adequate iron intakes that met the criteria of a nutrient adequacy score of ‘1’ for calculating the DRIs-score (Table 1). The proportion of women with adequate iron intakes was lower than that of men (*p* < 0.001: Table 1). The recommended intake of iron for premenopausal women is higher than that for men because of iron losses through menstruation. According to the 2017 National Health and Nutrition Survey in Japan, the iron intake of adult women was 7.5 mg/day [24], and the iron intake in the highest score group of the Prudent dietary pattern quartile was 9.95 mg/day in a large-scale study of Japanese dietary patterns [8]. Both of these intakes are below the RDA for iron of 10.5 mg/day for premenopausal women [25]. These results suggest that the iron intakes of premenopausal women are unlikely to reach recommended levels even when their healthy dietary pattern scores are high. Furthermore, the DRIs-J 2015 for vitamin D, vitamin K, vitamin B_12_, folic acid, and vitamin C are the same for both men and women. When the dietary reference intake values for micronutrients are calculated using the nutrient density method (per estimated energy requirement of 1000 kcal) [26], the reference values for women are higher than those for men. The proportions of participants with adequate intake of vitamin D, folic acid, and vitamin C for women were lower than those for men (Table 1). Thus, inadequate iron intakes in women and the difference in dietary reference intake values for the DRIs-score calculation between men and women may explain the reason why healthy dietary pattern scores were higher in women than in men, despite no significant difference in DRIs-scores between men and women.

We showed that healthy dietary pattern scores were strongly and positively associated with DRIs-scores in both men and women. Furthermore, a plateau of the DRIs-score at around the highest tertile of the healthy dietary pattern score was observed in both men and women. These results suggest that most men and women in the highest tertile group had adequate intakes of almost all micronutrients. However, among the micronutrients used for calculating the DRIs-score, only salt intake was not within the DRIs-J 2015 reference range (i.e., salt intake exceeded the DG) in the individuals with a high healthy dietary pattern score. In previous studies, healthy Japanese dietary patterns were positively related to salt intake [8,10,13], which is consistent with our findings. Interestingly, a previous study demonstrated that the Japanese healthy dietary pattern was negatively associated with systolic blood pressure, despite a high intake of salt [27], which suggests an antihypertensive effect of the Japanese healthy dietary pattern. The Japanese healthy dietary pattern, characterized by a high consumption of vegetables and fruits, is similar to the DASH diet [28], which is effective in preventing hypertension. Therefore, the antihypertensive effect of the Japanese healthy dietary pattern despite high intakes of salt may be due to a high consumption of vegetables and fruits. Nevertheless, a reduced intake of salt is recommended to prevent hypertension and stroke [29,30]; it should be noted that the Japanese healthy dietary pattern is not associated with an appropriate intake of salt, which is a factor that should be assessed in isolation from dietary pattern.

This study has some limitations. First, although we confirmed the validity of the nutrient and food intake amounts, they are taken from a BDHQ and are estimated values, and potential misclassification of dietary patterns may have affected our result. Therefore, further research using more accurate dietary information is needed. Second, the participants of this study were alumni of the same university, and their spouses, who wished to participate in the study; therefore, our results may be affected by self-selection bias. Thus, further investigations among representative populations are necessary to generalize our findings to the entire Japanese population. Third, not meeting the RDA does not necessarily mean a participant has an inadequate intake. Because RDAs are set at the 97th-98th percentile of requirements, some participants with intakes less than RDA may have intakes appropriate to their requirements. Thus, some of our participants could have been inappropriately classified.

## 5. Conclusions

The present study indicates that the Japanese healthy dietary pattern is associated with adequate micronutrient intakes in both men and women, based on the DRIs-J 2015. Our findings suggest that the Japanese healthy dietary pattern represents a balanced intake of multiple micronutrients in both men and women.

## Figures and Tables

**Figure 1 nutrients-12-00006-f001:**
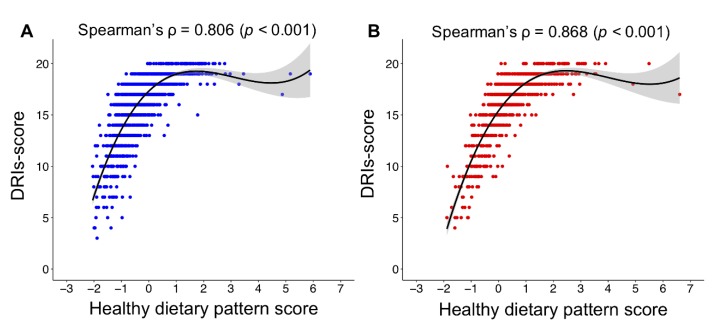
Relationship between the healthy dietary pattern score and the DRIs-score in men (**A**) and women (**B**). Cubic spline curves are shown as a solid line with the shaded area representing the 95% confidence intervals. DRIs-score, dietary reference intakes score.

**Table 1 nutrients-12-00006-t001:** Characteristics of the study participants.

Variables	Men		Women		*p* ^a^	*p* ^b^
*n*	1418		795			
Age, year	56.1 ± 10.2		51.8 ± 8.4		<0.001	
Body mass index, kg/m^2^	23.8 ± 3.0		21.4 ± 3.0		<0.001	
Healthy dietary pattern score (range)	−0.25 ± 0.87	(−2.05–5.89)	0.45 ± 1.06	(−1.89–6.61)	<0.001	
DRIs-score (range)	15.9 ± 3.1	(3–20)	15.9 ± 3.3	(4–20)	0.862	
**Energy and nutrient intake**						
Energy, kcal/day	2074 ± 541		1736 ± 454		<0.001	
Protein, % energy	15.4 ± 2.9	(74.3%)	16.6 ± 3.2	(77.4%)	<0.001	0.104
Fat, % energy	27.0 ± 5.7	(68.5%)	29.7 ± 5.5	(57.9%)	<0.001	<0.001
Carbohydrate, % energy	57.6 ± 7.8	(73.3%)	53.7 ± 7.6	(67.3%)	<0.001	0.003
**Micronutrients intakes**						
Vitamin A, μgRAE/1000 kcal ^c^	446.9 ± 218.4	(62.1%)	494.6 ± 250.7	(67.7%)	<0.001	0.009
Vitamin D, μg/1000 kcal	7.5 ± 4.3	(97.0%)	8.1 ± 4.9	(94.8%)	0.003	0.009
Vitamin E, mg/1000 kcal	4.1 ± 1.0	(94.5%)	4.7 ± 1.1	(95.1%)	<0.001	0.548
Vitamin K, μg/1000 kcal	181 ± 87.9	(97.7%)	214 ± 100	(96.7%)	<0.001	0.152
Vitamin B_1_, mg/1000 kcal	0.43 ± 0.09	(11.4%)	0.49 ± 0.10	(15.7%)	<0.001	0.004
Vitamin B_2_, mg/1000 kcal	0.75 ± 0.19	(78.4%)	0.83 ± 0.20	(89.9%)	<0.001	<0.001
Niacin, mgNE/1000 kcal ^d^	10.1 ± 2.4	(98.9%)	10.7 ± 2.6	(98.6%)	<0.001	0.495
Vitamin B_6_, mg/1000 kcal	0.72 ± 0.16	(83.9%)	0.80 ± 0.19	(85.4%)	<0.001	0.332
Vitamin B_12_, μg/1000 kcal	5.4 ± 2.5	(99.7%)	5.5 ± 2.7	(99.5%)	0.365	0.406
Folate, μg/1000 kcal	192 ± 61	(97.4%)	225 ± 78	(94.2%)	<0.001	<0.001
Pantothenic acid, mg/1000 kcal	3.65 ± 0.74	(99.5%)	3.99 ± 0.78	(99.0%)	<0.001	0.158
Vitamin C, mg/1000 kcal	63 ± 27.4	(82.5%)	78 ± 33	(79.1%)	<0.001	0.050
Sodium (salt-equivalent), g/1000 kcal	5.9 ± 1.1	(0.4%)	5.7 ± 1.1	(1.9%)	0.013	<0.001
Potassium, mg/1000 kcal	1435 ± 367	(72.5%)	1664 ± 424	(79.2%)	<0.001	<0.001
Calcium, mg/1000 kcal	297 ± 100	(54.5%)	350 ± 111	(50.3%)	<0.001	0.058
Magnesium, mg/1000 kcal	143 ± 30	(45.7%)	157 ± 34	(54.6%)	<0.001	<0.001
Phosphorus, mg/1000 kcal	583 ± 117	(96.2%)	637 ± 131	(98.5%)	<0.001	0.002
Iron, mg/1000 kcal	4.4 ± 1.0	(94.2%)	4.9 ± 1.2	(62.9%)	<0.001	<0.001
Zinc, mg/1000 kcal	4.4 ± 0.7	(71.4%)	4.7 ± 0.8	(83.5%)	<0.001	<0.001
Copper, mg/1000 kcal	0.61 ± 0.11	(98.0%)	0.66 ± 0.12	(98.5%)	<0.001	0.370
Manganese, mg/1000 kcal	1.65 ± 0.47	(48.6%)	1.76 ± 0.49	(41.8%)	<0.001	0.002

Data are mean ± standard deviation and percentage of nutrient intake that met or exceed the Dietary Reference Intakes for Japanese 2015. ^a^ Data were analyzed using Student’s *t* test; ^b^ Data were analyzed using Chi-square test; ^c^ 1 μgRAE = retinol (μg) + β-carotene (μg) × 1/12 + α-carotene (μg) × 1/24 + β-cryptoxanthin (μg) × 1/24 + other provitamin A carotenoids (μg) × 1/24; ^d^ 1 mgNE = niacin intake (mg) + (1/60) tryptophan intake (mg); DRIs-score, dietary reference intakes score; RAE, retinol activity equivalent; NE, niacin equivalent.

**Table 2 nutrients-12-00006-t002:** Nutrient intake according to the tertiles of the healthy dietary pattern scores in men.

Variables	Tertiles (T) of the Healthy Dietary Pattern Score						*p* ^a^	*p* ^b^
	T1 (Low)		T2 (Middle)		T3 (High)			
*n*	472		473		473			
Healthy dietary pattern score	−1.06 ± 0.36		−0.35 ±0.30		0.66 ± 0.76		<0.001	
DRIs-score	13.0 ± 3.0		16.3 ± 1.8		18.3 ± 1.3		<0.001	
**Energy and macronutrient intake**								
Energy, kcal/day	2075 ± 552		2103 ± 540		2046 ± 531		0.412	
Protein, % energy	13.4 ± 2.0	(55.5%)	15.2 ± 1.9	(88.4%)	17.6 ± 3.1	(78.9%)	<0.001	<0.001
Fat, % energy	24.1 ± 5.3	(69.3%)	27.4 ± 4.9	(73.6%)	29.5 ± 5.5	(62.6%)	<0.001	0.027
Carbohydrate, % energy	62.5 ± 6.6	(68.4%)	57.4 ± 6.0	(82.5%)	52.9 ± 7.5	(69.1%)	<0.001	0.810
**Micronutrients intakes**								
Vitamin A, μgRAE/1000 kcal ^c^	347 ± 209	(42.6%)	448 ± 200	(61.7%)	545 ± 200	(82.0%)	<0.001	<0.001
Vitamin D, μg/1000 kcal	5.5 ± 2.8	(94.3%)	7.3 ± 3.4	(98.1%)	9.7 ± 5.2	(98.7%)	<0.001	<0.001
Vitamin E, mg/1000 kcal	3.3 ± 0.6	(84.5%)	4.1 ± 0.6	(98.9%)	4.9 ± 0.9	(100%)	<0.001	<0.001
Vitamin K, μg/1000 kcal	126 ± 53	(93.4%)	171 ± 61	(99.8%)	246 ± 96	(100%)	<0.001	<0.001
Vitamin B_1_, mg/1000 kcal	0.35 ± 0.06	(0.4%)	0.42 ± 0.05	(2.5%)	0.51 ± 0.08	(31.3%)	<0.001	<0.001
Vitamin B_2_, mg/1000 kcal	0.62 ± 0.14	(52.5%)	0.74 ± 0.13	(87.1%)	0.88 ± 0.18	(95.6%)	<0.001	<0.001
Niacin, mgNE/1000 kcal ^d^	8.8 ± 1.9	(96.8%)	9.9 ± 1.9	(100%)	11.5 ± 2.5	(100%)	<0.001	<0.001
Vitamin B_6_, mg/1000 kcal	0.60 ± 0.11	(60.4%)	0.70 ± 0.10	(91.3%)	0.87 ± 0.15	(99.8%)	<0.001	<0.001
Vitamin B_12_, μg/1000 kcal	4.3 ± 1.9	(99.2%)	5.4 ± 2.1	(100%)	6.6 ± 2.8	(100%)	<0.001	0.014
Folate, μg/1000 kcal	145 ± 34	(94.1%)	185 ± 35	(100%)	245 ± 61	(98.1%)	<0.001	<0.001
Pantothenic acid, mg/1000 kcal	3.13 ± 0.55	(98.5%)	3.59 ± 0.49	(100%)	4.22 ± 0.71	(100%)	<0.001	0.001
Vitamin C, mg/1000 kcal	45 ± 18	(57.0%)	60 ± 17	(91.5%)	84 ± 29	(98.9%)	<0.001	<0.001
Sodium (salt-equivalent), g/1000 kcal	5.5 ± 1.1	(0.6%)	5.8 ± 1.0	(0.4%)	6.3 ± 1.2	(0%)	<0.001	0.099
Potassium, mg/1000 kcal	1133 ± 206	(34.1%)	1387 ± 191	(83.7%)	1784 ± 331	(99.6%)	<0.001	<0.001
Calcium, mg/1000 kcal	233 ± 75	(25.2%)	289 ± 71	(54.1%)	369 ± 100	(84.1%)	<0.001	<0.001
Magnesium, mg/1000 kcal	121 ± 18	(11.7%)	140 ± 18	(41.9%)	169 ± 29	(83.5%)	<0.001	<0.001
Phosphorus, mg/1000 kcal	498 ± 78	(89.8%)	574 ± 74	(99.2%)	676 ± 117	(99.6%)	<0.001	<0.001
Iron, mg/1000 kcal	3.6 ± 0.6	(83.1%)	4.3 ± 0.6	(99.6%)	5.3 ± 1.0	(100%)	<0.001	<0.001
Zinc, mg/1000 kcal	3.9 ± 0.6	(44.7%)	4.4 ± 0.5	(77.8%)	4.9 ± 0.6	(91.8%)	<0.001	<0.001
Copper, mg/1000 kcal	0.54 ± 0.09	(94.3%)	0.60 ± 0.08	(99.6%)	0.69 ± 0.11	(100%)	<0.001	<0.001
Manganese, mg/1000 kcal	1.53 ± 0.45	(37.5%)	1.64 ± 0.44	(46.1%)	1.80 ± 0.48	(62.2%)	<0.001	<0.001

Data are mean ± standard deviation and percentage of nutrient intake that met or exceed the Dietary Reference Intakes for Japanese 2015. ^a^ Data were analyzed using linear regression (for continuous variables); ^b^ Data were analyzed using Mantel-Haenszel Chi-square test (for categorical variables); ^c^ 1 μgRAE = retinol (μg) + β-carotene (μg) × 1/12 + α -carotene (μg) × 1/24 + β -cryptoxanthin (μg) × 1/24 + other provitamin A carotenoids (μg) × 1/24; ^d^ 1 mgNE = niacin intake (mg) + (1/60) tryptophan intake (mg); DRIs-score, dietary reference intakes score; RAE, retinol activity equivalent; NE, niacin equivalent.

**Table 3 nutrients-12-00006-t003:** Nutrient intake according to the tertiles of the healthy dietary pattern scores in women.

Variables	Tertiles (T) of the Healthy Dietary Pattern Score						*p* ^a^	*p* _b_
	T1 (Low)		T2 (Middle)		T3 (High)			
*n*	265		266		264			
Healthy dietary pattern score	−0.56 ± 0.45		0.33 ± 0.38		1.57 ± 0.84		<0.001	
DRIs-score	12.6 ± 3.1		16.6 ± 1.8		18.4 ± 1.3		<0.001	
**Energy and macronutrient intake**								
Energy, kcal/day	1688 ± 459		1769 ± 417		1752 ± 482		0.107	
Protein, % energy	14.3 ± 2.0	(77.0%)	16.6 ± 2.3	(89.5%)	18.9 ± 3.3	(65.5%)	<0.001	0.002
Fat, % energy	27.6 ± 5.1	(69.4%)	30.2 ± 4.8	(55.6%)	31.4 ± 5.7	(48.5%)	<0.001	<0.001
Carbohydrate, % energy	58.1 ± 6.3	(79.6%)	53.1 ± 6.3	(71.8%)	49.7 ± 7.7	(50.4%)	<0.001	<0.001
**Micronutrients intakes**								
Vitamin A, μgRAE/1000 kcal ^c^	356 ± 160	(35.8%)	477 ± 202	(73.3%)	651 ± 281	(93.9%)	<0.001	<0.001
Vitamin D, μg/1000 kcal	5.9 ± 3.3	(89.1%)	8.1 ± 4.2	(96.2%)	10.3 ± 5.8	(99.2%)	<0.001	<0.001
Vitamin E, mg/1000 kcal	3.8 ± 0.7	(86.4%)	4.7 ± 0.7	(98.9%)	5.6 ± 1.0	(100%)	<0.001	<0.001
Vitamin K, μg/1000 kcal	144 ± 60	(90.6%)	198 ± 62	(99.6%)	299 ± 100	(100%)	<0.001	<0.001
Vitamin B_1_, mg/1000 kcal	0.40 ± 0.07	(0.8%)	0.49 ± 0.07	(6.0%)	0.58 ± 0.08	(40.5%)	<0.001	<0.001
Vitamin B_2_, mg/1000 kcal	0.69 ± 0.14	(74.0%)	0.82 ± 0.15	(96.2%)	0.97 ± 0.19	(99.6%)	<0.001	<0.001
Niacin, mgNE/1000 kcal ^d^	8.9 ± 1.9	(95.8%)	10.8 ± 2.1	(100%)	12.3 ± 2.7	(100%)	<0.001	<0.001
Vitamin B_6_, mg/1000 kcal	0.63 ± 0.11	(59.2%)	0.79 ± 0.11	(97.4%)	0.98 ± 0.16	(99.6%)	<0.001	<0.001
Vitamin B_12_, μg/1000 kcal	4.4 ± 2.0	(98.9%)	5.6 ± 2.4	(99.6%)	6.7 ± 3.2	(100%)	<0.001	0.066
Folate, μg/1000 kcal	161 ± 37	(86.0%)	217 ± 42	(98.9%)	298 ± 76	(97.7%)	<0.001	<0.001
Pantothenic acid, mg/1000 kcal	3.39 ± 0.54	(97.0%)	3.96 ± 0.55	(100%)	4.62 ± 0.69	(100%)	<0.001	0.001
Vitamin C, mg/1000 kcal	52 ± 20	(46.4%)	77 ± 21	(91.7%)	104 ± 32	(99.2%)	<0.001	<0.001
Sodium (salt-equivalent), g/1000 kcal	5.3 ± 0.9	(4.2%)	5.8 ± 1.0	(0.4%)	6.1 ± 1.3	(1.1%)	<0.001	0.011
Potassium, mg/1000 kcal	1282 ± 244	(41.5%)	1643 ± 212	(96.2%)	2070 ± 351	(100%)	<0.001	<0.001
Calcium, mg/1000 kcal	276 ± 85	(17.4%)	345 ± 80	(51.9%)	431 ± 106	(81.8%)	<0.001	<0.001
Magnesium, mg/1000 kcal	128 ± 19	(10.6%)	154 ± 19	(59.4%)	188 ± 31	(93.9%)	<0.001	<0.001
Phosphorus, mg/1000 kcal	540 ± 82	(95.8%)	634 ± 90	(100%)	738 ± 132	(99.6%)	<0.001	<0.001
Iron, mg/1000 kcal	3.9 ± 0.7	(45.7%)	4.9 ± 0.7	(59.4%)	6.1 ± 1.1	(83.7%)	<0.001	<0.001
Zinc, mg/1000 kcal	4.2 ± 0.6	(64.2%)	4.7 ± 0.6	(88.3%)	5.3 ± 0.7	(98.1%)	<0.001	<0.001
Copper, mg/1000 kcal	0.57 ± 0.09	(95.8%)	0.65 ± 0.08	(99.6%)	0.75 ± 0.10	(100%)	<0.001	<0.001
Manganese, mg/1000 kcal	1.59 ± 0.44	(25.7%)	1.80 ± 0.46	(47.7%)	1.91 ± 0.53	(51.9%)	<0.001	<0.001

Data are mean ± standard deviation and percentage of nutrient intake that met or exceed the Dietary Reference Intakes for Japanese 2015. ^a^ Data were analyzed using linear regression (for continuous variables); ^b^ Data were analyzed using Mantel-Haenszel Chi-square test (for categorical variables); ^c^ 1 μgRAE = retinol (μg) + β-carotene (μg) × 1/12 + α -carotene (μg) × 1/24 + β -cryptoxanthin (μg) × 1/24 + other provitamin A carotenoids (μg) × 1/24; ^d^ 1 mgNE = niacin intake (mg) + (1/60) tryptophan intake (mg); DRIs-score, dietary reference intakes score; RAE, retinol activity equivalent; NE, niacin equivalent.

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
