# Peer review of "Micronutrient Intake Adequacy in Men and Women with a Healthy Japanese Dietary Pattern"

_nutrients, 2019, doi:10.3390/nu12010006_

Round 1
Reviewer 1 Report
Thank you for answer all my questions. I just have an additional suggestion.
Although the main aim of the paper was to compare the healthy dietary pattern with micronutrient intake, could you say something in results/discussion regarding the macronutrients adequacy? (since you have included that information in the tables). It seems from the tables (2/3) that the middle tertile is the one with a better macronutrient profile.
Reviewer 2 Report
The authors answer insists on the interest of the article, and I still have some doubts about its novelty and impact on the current knowledge of the field. Indeed, the relationship between healthy diet and nutrient intake has been full established.
In any case, the limitations of the study, which authors describe perfectly, and that I detailed in my comments to the editor on R1, make the results of little relevance.
Reviewer 3 Report
As this is a re-review, I believe the authors have improved the manuscript since the first reading/editing.
The track changes probably should have been accepted before re-submitting, but that is perhaps only my personal opinion.
However, the information is interesting and will be useful to certain nutritional groups. I hope the authors continue to do follow-up research....and perhaps find opportunities to educate the masses regarding components of a truly healthy and balanced diet.
Reviewer 4 Report
The Authors have addressed all my comments
Author Response
Please see the attachment.

This manuscript is a resubmission of an earlier submission. The following is a list of the peer review reports and author responses from that submission.
Round 1
Reviewer 1 Report
The manuscript by Ito and colleagues aims to examine the relationship between a healthy Japanese dietary pattern and micronutrient intake adequacy based on the Dietary Reference Intakes for Japanese 2015 (DRIs-J 2015). This manuscript is well-written and well-conducted, however, I have some doubts about its novelty and impact on the current knowledge of the field. Indeed, the relationship between healthy diet and nutrient intake has been full established.
Moreover, I don't understand why the Authors used only the first dietary pattern derived by the principal component analysis. My suggestion is to compare nutrient intake across different dietary patterns. Otherwise, I believe that this manuscript, in the current version, is not suitable for adding novel evidence to the research.
Reviewer 2 Report
This is an interesting study in the sense that the sample size was reasonable, and that both men and women were tested. You did not mention anything about food bioavailability though, and while it may seem obvious that nutrients must have been bioavailable, based upon the final results, this aspect could still be responsible for some of the differences between groups. It is difficult to know for certain...but it would be prudent to include a paragraph or so regarding the bioavailability issue.
A sample of a typical daily menu would be informative for the reader as well, but perhaps not essential for the purpose of this article.
Reviewer 3 Report
The article is not justified by the novelty (it is not very original), nor by the interest of the results that shows.It would be of more interest if, in the discussion, they had commented on the correlation between the intake of foods with the results obtained for the intake of micronutrients. In any case, the study has important limitations that they are perfectly specified and that, in my opinion, make the results of little relevance
Reviewer 4 Report
Thank you for giving me an opportunity to review this manuscript. The study compare the micronutrient intake and adequacy by tertiles of a healthy dietary pattern obtained by principal component analysis. I have included below some comments/suggestions.
Abstract
Please add in the abstract information on the dietary method used to collect participates’ dietary information
Methods
2.1 . Please add the selection/exclusion criteria for the participants
The authors mention the possibility of underestimation (line 119.) Have the authors evaluated the misreporting in the sample?
The Healthy dietary pattern score was compared with micronutrient intake assesses from the same questionnaire – so it was expected that the individuals with this dietary intake to have better dietary adequacy. Shouldn’t the authors compare the nutrient intake assess by another independent dietary method (e.g. dietary records)?
Please add the range of the DRI-score and the dietary pattern score (in methods or results)
Results/discussion
Please add information of adequacy of macronutrients by tertiles of healthy dietary pattern (table 3)
